# Naringenin Prevents Oxidative Stress and Inflammation in LPS-Induced Liver Injury through the Regulation of LncRNA-mRNA in Male Mice

**DOI:** 10.3390/molecules28010198

**Published:** 2022-12-26

**Authors:** Mengting Ji, Zhao Deng, Xiaoyin Rong, Ruixiao Li, Ziwei You, Xiaohong Guo, Chunbo Cai, Yan Zhao, Pengfei Gao, Guoqing Cao, Bugao Li, Yang Yang

**Affiliations:** 1College of Animal Science, Shanxi Agricultural University, Taigu 030801, China; 2Institute of Genetics and Developmental Biology, Chinese Academy of Sciences, Beijing 100101, China

**Keywords:** naringenin, liver, oxidative stress, inflammation, RNA-seq, lncRNA

## Abstract

Inflammation accompanies hepatic dysfunction resulting from tissue oxidative damage. Naringenin (Nar), a natural flavanone, has known antioxidant and anti-inflammatory activities, but its mechanism of action in the regulation of liver dysfunction requires further investigation. In this study, the role of naringenin in lipopolysaccharide (LPS)-induced hepatic oxidative stress and inflammation was explored, as well as its mechanism by transcriptome sequencing. The results indicated that compared with the LPS group, Nar treatment caused a significant increase in the mRNA levels of antioxidant factors glutamate-cysteine ligase catalytic subunit (GCLC) and glutamate-cysteine ligase modifier subunit (GCLM), yet the expression of related inflammatory factors (MCP1, TNFα, IL-1β and IL-6) showed less of an increase. RNA sequencing identified 36 differentially expressed lncRNAs and 603 differentially expressed mRNAs. KEGG enrichment analysis indicated that oxidative stress and inflammation pathways are meticulously linked with naringenin treatment. The Co-lncRNA-mRNA network was also constructed. Tissue expression profiles showed that lncRNA played a higher role in the liver. Subsequently, expression levels of inflammatory factors indicated that lncRNAs and target mRNAs were significantly reduced after naringenin treatment in mouse liver AML12 cells and obese mouse. These results suggest that naringenin helps to prevent liver dysfunction through the regulation of lncRNA-mRNA axis to reduce oxidative stress and inflammatory factors.

## 1. Introduction

The liver maintains numerous vital functions, including detoxification, metabolism, protein synthesis, host defense, and immune response [1]. Recent reports have shown the binding ability of lipopolysaccharide (LPS) to the corresponding binding protein. It then connects with toll-like receptor 4 (TLR4) after entering the liver. This has led to the activation of various cellular reaction networks and NF-κB pathways [2]. Liver toxins initially damage the central lobule area of the liver, releasing reactive oxygen species (ROS), inducing lipid peroxidation and the production of pro-inflammatory cytokines, such as tumor necrosis factor-α (TNFα), interleukin-1β (IL-1β), and interleukin 6 (IL-6) [3]. In response, hepatocytes rapidly uptake LPS, which is followed by inflammatory response, oxidative stress, and hepatic necrosis, which contribute to the development of LPS-induced acute liver injury. Notably, TRIM protein family members have been linked to the regulation of innate immunity [4,5,6]. Specifically, TRIM protein promotes host immune response through the ubiquitination of key signal molecules in immunity, and controls pathogen infection by regulating the signal pathway of pattern recognition receptor [7]. Trim56 is related to the regulation of innate immune responses caused by double-stranded DNA and several viral pathogens [8,9]. However, there is no specific report on how liver injury affects regulation of the Trim family in the literature. In recent years, an increasing number of studies have focused on the hepatoprotective properties of natural products, such as quercetin, berberine, and curcumin [10,11,12]. Published reports have suggested that flavonoids have antioxidant and anti-inflammatory properties, which could alleviate liver damage [13]. The potent antioxidant flavonoids of quercetin and anthocyanins, as well as the non-flavonoid polyphenol resveratrol, have been shown to reduce ROS levels [14,15,16,17]. Naringenin (Nar) has attracted considerable attention due to its ability to combat lipid peroxidation. Miler et al. reported an improvement in the antioxidant status and membrane lipid compositions in the liver of old-aged Wistar rats following Nar administration [18]. Meanwhile, Nar has been shown to reduce lipid peroxidation in oxytetracycline-induced liver oxidative stress [19]. Elsewhere, evidence from both in vivo and in vitro studies has demonstrated the therapeutic potential of the botanic compound Nar in the treatment of obesity and diabetes [13,20].

Nar, or 4′,5,7-trihydroxyflavanone, is a flavonoid that is found in abundance in citrus fruits such as lemons and oranges [21]. It is a key component in traditional Chinese medicine due to its pharmacological actions, including anti-inflammatory, antioxidant, anti-lipogenic, anti-atherosclerotic, and hepatoprotective effects [22,23]. Its biological activity derives from its structure–activity relationship. Its hydroxyl group provides hydrogen to ROS, which can then maintain a stable structure, thereby demonstrating the free radical scavenging ability of Nar [24,25]. The prevention of oxidative stress, promotion of fibrogenesis, and inhibition of the trans-differentiation of hepatic stellate cells are the main hepatoprotective effects of Nar [26]. Moreover, researchers have demonstrated that Nar can induce hepatic fatty acid oxidation, reduce lipid utilization, normalize hepatic triglyceride (TG) and blood glucose levels, reduce hepatic lipid accumulation, and improve dyslipidemia, as well as help to correct many metabolic disorders associated with ischemia reperfusion (IR) [27,28]. In addition, in doxorubicin-induced rat liver injury, Nar treatment was shown to have protective effects against hepatocyte necrosis, cholestasis, and membrane permeability, whilst also significantly reducing levels of the liver injury markers of aspartate aminotransferase (AST) and alanine aminotransferase (ALT) in mice [29]. Some studies have also shown that AST, ALT, and superoxide dismutase do not change significantly, but Nar treatment (50 mg/kg) effectively reduced oxytetracycline-induced oxidative damage in rat livers [19]. Other studies have reported the in vitro anti-inflammatory effect of Nar on adipocytes and high-fat diet-induced obese mouse adipose tissue [30,31].

Long non-coding RNAs (lncRNAs) are a series of single-stranded RNAs with a length of more than 200 nucleotides, which have minimal or no ability to encode proteins, but are involved in complex biological processes and pathophysiological conditions, including lipid metabolism disorders [32,33,34]. Clinical studies have shown that lncRNAs are crucial regulators in various diseases and can damage cholesterol in vivo. Moreover, they maintain a critical role in the progression of lipid-related diseases [35,36,37]. Although the specific roles of lncRNAs in liver injury remain undetermined, they have been established as vital regulators of gene expression related to metabolic homeostasis or dysfunction. For example, lncRNAs undergo regulation during adipogenesis and inflammation, with many being bound to their promoters by key transcription factors, such as PPARγ, CEBPα, TNFα, and IL-6 [38,39]. Furthermore, considerable evidence highlights the association of liver diseases with abnormal expression of lncRNAs [40,41,42].

The objective of our work was to evaluate the potential of naringenin in the prevention of hepatic injury induced by LPS in mice. The current study investigated changes in liver-related morphological structures, as well as inflammatory and antioxidant factors, after LPS treatment. Subsequently, differentially expressed lncRNAs (DElncRNAs) and mRNAs were detected by RNA-seq, and a related network was constructed to explain the mechanism of action of Nar.

## 2. Results

### 2.1. Nar Alters LPS-Induced Changes in Serum Levels

To investigate the effect of Nar on LPS-induced liver injury in mice, we detected different indexes in the serum of mice. The results show that compared with the control (CTRL) group, serum glucose and TG levels of the LPS-treated group decreased by 25.72% and 21.78%, respectively. However, both TG and glucose showed an increase in the LPS+Nar (L_Nar) treatment group compared with the LPS group (Figure 1a,b). To further investigate the preventive effect of Nar on LPS-induced liver injury, the levels of AST and ALT were also determined. The results revealed that ALT levels were significantly increased in the LPS group compared to the CTRL group (*p* < 0.01). In contrast, the L_Nar group reduced the LPS-induced increase in ALT (Figure 1c). Similarly, AST levels also showed a decrease in the L_Nar group compared to the LPS group (Figure 1d).

### 2.2. Nar Prevented LPS-Induced Inflammation and Oxidative Stress

Histological analysis showed LPS-induced inflammatory infiltration of hepatocytes, separation of hepatic sinuses, and periportal fibrosis. However, these were reduced in the L_Nar group compared with the LPS group (Figure 2a).

We also examined the effect of Nar on the pro-inflammatory gene expression of mice with liver damage. This was achieved by measuring MCP1, TNFα and IL-6 mRNA levels by qRT-PCR (Figure 2b). The results showed that the mRNA levels of MCP1, TNFα and IL-6 were significantly higher in the LPS group (*p* < 0.01). However, the L_Nar group inhibited an increase compared to the LPS group. Furthermore, the mRNA levels of the antioxidant factors GCLC and GCLM in the L_Nar group were also significantly higher than those in the LPS group (*p* < 0.01), indicating that they had higher antioxidant capacity (Figure 2c).

### 2.3. Nar Induced Transcriptome Alterations in the Liver

DElncRNAs in the liver were further analyzed using RNA-seq. In the CTRL vs LPS comparison group, there were 640 DElncRNAs, of which 275 were upregulated and 265 were downregulated. In the group LPS vs L_Nar, we found 36 DElncRNAs, of which 18 were downregulated and 18 upregulated (Appendix A). Meanwhile, there were 14 identical DElncRNAs between the two groups (Figure 3a–c). The principle component analysis (PCA) is shown in Figure 3d. The PC1 value is 66.62% and the PC2 value is 10.75%. Hierarchical cluster analysis demonstrated that, compared with the LPS group, part of the DELncRNAs in the L_Nar group showed an opposite trend in the LPS group (Figure 3e). Among them, the expression of two DELncRNAs was downregulated and then upregulated, while the expression of four DELncRNAs was first upregulated and then downregulated (Figure 3f).

### 2.4. Nar Changed the Expression of DEmRNAs in Mice Liver

A comparison of the transcriptomic changes between CTRL and LPS groups revealed 4908 differentially expressed mRNAs (DEmRNAs), which included 2770 upregulated and 2138 downregulated DEmRNAs. The comparison results of the LPS and L_Nar groups exhibited 603 DEmRNAs, including 373 upregulated and 230 downregulated DEmRNAs (Appendix A), while the comparison of the CTRL and L_Nar groups identified 8110 DEmRNAs, including 4390 upregulated and 3720 downregulated DEmRNAs (Figure 4a,b).

Hierarchical clustering analysis revealed the altered expression of DEGs following Nar treatment in the LPS group (Figure 4c, Appendix A). The KEGG classification results indicated that the KEGG terms annotated via DEmRNA source genes in the LPS and L_Nar groups involved three functional classifications, lipid metabolism, inflammation-related signaling pathways, and bile secretion (Figure 4d, Appendix A). However, Nar treatment appears to be associated with signaling pathways related to inflammation and lipid metabolism, such as the NOD-like receptor signaling pathway, the PPAR signaling pathway, and retinol metabolism. 

### 2.5. LncRNA and mRNA Differential Gene Co-Expression Networks in the Liver

RNA-seq identified 36 DElncRNAs and 603 DEmRNAs. Seventeen DElncRNAs were matched and interaction plots were constructed with their corresponding DEmRNAs (Figure 5, Appendix A). Fourteen of them can be presented in a network. Interestingly, S100a8 and S100a9 have been shown to play a central role in the inflammatory process [43]. In this network, we found that this gene was connected with MSTRG.6826.5 and MSTRG.19540.3, so we infer that the lncRNA may play a regulatory role in the regulation of liver damage by naringenin.

### 2.6. Nar Prevents the Expression of lncRNA and Its Target Genes

The protein coding abilities of MSTRG.19540.3 and MSTRG.6826.5 were predicted and analyzed through CPAT and CPC online websites, and the reported non-coding gene HOTAIR was selected as a reference. The results showed that the lncRNA sequence did not have the ability to code protein (Figure 6a). To clarify the tissue expression characteristics of lncRNA, the expressions of MSTRG.19540.3 and MSTRG.6826.5 in the heart, liver, spleen, lung, fat, leg muscles, and colon of mice were detected by qRT-PCR, respectively. Results showed that it was expressed in all tissues, but the highest expression level was found in liver tissue (Figure 6b). Further, the function of lncRNA and its target gene were verified in AML-12 cells. The results showed that the expression of lncRNA and its target genes significantly increased in the LPS group compared with the NC group (*p* < 0.05). Compared with the LPS group, the expression of lncRNA and its target genes in the L_Nar group was significantly inhibited (*p* < 0.05). In addition, naringenin alone can also inhibit its expression (Figure 6c). Similarly, in obese mice induced by diet, naringenin treatment has the same therapeutic effects on lnRNA and its target genes (Figure 6d).

### 2.7. Nar Reduces Liver Inflammatory in AML-12 Cells and Diet-Induced Obese Mice

A low-fat diet (LFD)/high-fat diet (HFD)-fed obese mouse model was used to explore the effect of Nar on AML-12 cells and obesity-induced liver inflammation. The results indicated that the mRNA expression levels of IL-6, IL-1β, MCP1, and TNFα were significantly decreased in both the L_Nar group and Nar group compared with the LPS group (*p* < 0.01) (Figure 7a). Moreover, Nar also reduced the expression levels of inflammatory factors in diet-induced obese mice (Figure 7b). Through phenotype data, we found that Nar can prevent LPS-induced hepatotoxicity in mice. Then, RNA-seq was sequenced to construct a co-expression network of lncRNAs and mRNAs. The preventive effect of Nar was verified in the liver of diet induced obese mice and in AML-12 cells. The experimental schematic diagram is shown in Figure 7c.

## 3. Discussion

The liver is vital for maintaining homeostasis, and liver dysfunction is also strongly associated with oxidative stress and inflammatory response [44,45,46]. Published reports show that naringenin has a wide range of pharmacological activities, but its potential mechanisms in the liver still need to be determined [47,48]. In this study, naringenin, a new flavonoid compound in recent years, was used as the experimental object to investigate its preventive effect on LPS-induced liver dysfunction. Our results show that mice in the LPS group exhibit a series of significant symptoms of liver inflammatory infiltration and oxidative stress compared to the control group. Specifically, the expression of liver damage-specific factors (AST and ALT) is increased and the expression of antioxidant factors (GCLC, GCLM) is reduced, indicating that naringenin treatment has a certain therapeutic effect on liver damage [49]. Compared with the LPS group, naringenin treatment could effectively improve LPS-induced dyslipidemia in mice, prevent the expression of ALT and AST from rising, and increase the mRNA levels of GCLM and GCLC, indicating that naringenin treatment has a certain therapeutic effect on liver injury. Meanwhile, studies have found that the production of inflammatory cytokines and chemokines can result in hepatitis inflammation and chronic hepatitis [50,51]. Moreover, we found that inflammatory factors such as IL-1β, IL-6, MCP-1 and TNF-α decreased significantly after naringenin treatment compared with the LPS group. These results revealed that naringenin could relieve LPS-induced oxidative stress and liver inflammation.

Correlations between lncRNA expression patterns and mRNA expression patterns often occur at the transcriptional and post-transcriptional levels of cis, trans, or antisense regulators. This means that certain some lncRNAs may be co-regulated in expression networks [52]. Further, lncRNAs that act as scaffolds, decoys, or signals can function through genome targeting, cis- or trans-regulation, and antisense interference [53]. At present, the research focuses on Nar, a flavonoid with potential therapeutic effect in the treatment of LPS-induced liver injury. Specifically, Nar stimulates or modifies the expression of lncRNAs or mRNAs in the liver, thereby acting on the target genes to achieve the transmission of regulatory signals. In recent years, the development of high-throughput sequencing technology and bioinformatics analysis methods have resulted in an increase in the numbers of predicted and identified lncRNAs. In this study, 36 DElncRNAs and 603 DEmRNAs were identified by gene expression profiling. Moreover, this study extracted a group of LPS-regulated lncRNAs whose expression was reversed by Nar, including MSTRG.19540.3 and MSTRG.6826.5. Further functional identification showed that the lncRNA was significantly highly expressed in mouse liver tissues, indicating that MSTRG19540.3 and MSTRG.6826.5 may be a key factor for naringenin to affect liver dysfunction. Based on the KEGG annotation and the official classification, further evaluation of the signaling pathway enrichment analysis was conducted in terms of DEGs, which were identified through comparison of the Nar group with the LPS group. Studies have shown that LPS induces a series of metabolic disorders, mainly involving lipid metabolism, bile acid metabolism, and inflammatory signals. Furthermore, we found that inflammatory factors such as IL-1β, IL-6, MCP-1 and TNF-α decreased significantly after naringenin treatment compared with LPS group, indicating that the therapeutic effect of Nar may be effective in these signaling pathways and alleviate liver dysfunction by reducing the expression of its related factors [54].

In the analysis of co-expression sub-networks, the expression of MSTRG.19540.3 and MSTRG.6826.5 was shown to be associated with Trim56, Sl00a9, and Sl00a8. Amongst them, Trim56 is a member of the TRIM family of E3 ubiquitin ligases that have been implicated in a variety of functions. Kamanova et al. reported that the effector protein SopA operates by targeting the host E3 ubiquitin ligases Trim56 and Trim65, which have the ability to enhance interferon-β expression through the innate immune receptors RIG-I and MDA5 [55]. In the present study, we examined the expression of MSTRG.19540.3, MSTRG.6826.5 and their target genes (Trim56, Sl00a8, and Sl00a9) and found that Nar treatment prevented an increase in their expression. Of note, calcium-binding protein a8 (S100a8) and calcium-binding protein a9 (S100a9) are Ca^2+^ binding proteins belonging to S100 family, which can be used as sensors to express constitutive in neutrophils and monocytes [56]. At the same time, S100a8/a9 is released in inflammatory response and plays a key role in regulating the inflammatory response by stimulating leukocyte recruitment and inducing cytokine secretion [57]. The current study found that the mRNA expression levels of Sl00a8 and Sl00a9 were significantly lower in the Nar treatment group. Therefore, Nar may have anti-inflammatory action by preventing enhanced expression of Sl00a8 and Sl00a9 in LPS-induced liver injury. We posit that the inhibition of hepatic inflammation by Nar may be linked to the regulation between two specific lncRNAs (MSTRG.19540.3 and MSTRG.6826.5) and their target genes (Sl00a8, Sl00a9, and Trim56). However, the regulatory mechanism remains unclear, and thus requires further experimental exploration. Studies have found that a high-fat diet can induce obesity in mice and lead to nonalcoholic fatty liver disease [58,59]. This study also observed a marked decrease in the mRNA levels of the screened lncRNAs and their target genes in the liver of LFD/HFD-induced obese mice after Nar treatment, suggesting that this gene may also mitigate liver damage caused by obesity in mice. These findings have verified Nar as a promising candidate to treat obesity-related metabolic disorders. A further study of HFD-fed mice would offer mechanistic insights into liver damage.

## 4. Materials and Methods

### 4.1. Animals and Sample Collection

Eighteen 8-week-old male C57BL/6 mice (Speifu Biotechnology Co., Ltd, Beijing, China) received one week of adaptive feeding, before being randomly divided into three groups (*n* = 6), group I (CTRL group), group II (LPS group), and group III (L_Nar group). After Nar was dissolved in DMSO, it was diluted to a concentration of 30 mg/kg with PBS and administered to the third group of experimenters by gavage every day for one week, while groups I and II received intragastric administration the same doses of PBS for one week. On the eighth day, the mice in group II and group III received an intraperitoneal injection of 20 mg/kg LPS; the mice in group I were injected with 20 mg/kg of PBS. The subjects in all three groups were then sacrificed after eight hours.

Twenty-four other mice from the same batch as previously described were randomly divided into four groups (*n* = 6) after a feeding period of one week, group A (low-fat diet group), group B (low-fat diet + Nar group), group C (high-fat diet group), group D (high-fat diet + Nar group). Groups A and B received a daily low-fat diet (10% of calories from fat, D12450B, Beijing, China), while groups C and D received a daily high-fat diet (60% of calories from fat, D12492, Beijing, China). Concurrently, groups B and D received Nar (100 mg/kg/day) suspension by gavage for eight weeks, while groups A and C received the same volume of buffer solution by gavage for eight weeks. The mice were housed in a controlled environment at standard temperature (24 ± 2 °C) and humidity (45%) with a 12-hour light-dark cycle and had access to food and water throughout the experimental period. The mice were then euthanized and blood samples were collected. Additionally, a portion of the liver was taken from each subject and fixed in 4% paraformaldehyde, while another portion was taken and washed with PBS before being immediately immersed in RNAwait preservation solution (Catalogue SR0020, Solarbio Life Sciences, Beijing, China). These samples were stored at −80 °C. The remainder of the tissue was snap-frozen in liquid nitrogen and stored at −80 °C for subsequent analysis. This study followed the standards of the Animal Care and Use Committee of Shanxi Agricultural University (SXAU-EAW-2021MS.P.052801).

### 4.2. Cell Culture

The mouse liver cell line AML-12 was purchased from the Shanghai Institute of Biochemistry and Cell Biology, Chinese Academy of Sciences (Shanghai, China), and cultured in DMEM containing 10% fetal bovine serum (FBS, Gibco, ThermoFisher, Carlsbad, CA, USA). In the experiment, the seventh-tenth generation mouse hepatocyte line AML-12 was used. AML-12 cells in logarithmic growth phase were inoculated into six-well plates. After the cells reached a density of 70%–80%, they were treated with NC, LPS (200 ng/mL) and Nar (50 μg/mL) and LPS+Nar (L_Nar) treatment, and the cells are collected after 12 h for later analysis.

### 4.3. Enzyme-Linked Immunosorbent Assay and Biochemical Measurements

Commercial kits were purchased from Shanghai MLBio Shiye (Shanghai, China) to measure the levels of glucose (ml006368), TG (ml076637), AST (ml058659), and ALT (ml063179) in serum samples taken from the experimental subjects.

### 4.4. Histological Evaluation

Liver samples were kept in 4% paraformaldehyde (Solarbio Life Sciences, Beijing, China) at 4 °C overnight for subsequent histopathological analysis. The fixed livers were then dehydrated using ASP-200 (Leica, Berlin, Germany), embedded in paraffin, and sectioned at 5 μm thickness using a rotary microtome (RM2235, Leica, Berlin, Germany), as described in previously reported protocols [60]. Consecutive tissue sections were stained with haematoxylin-eosin and imaged microscopically (EVOS FL Auto, Life Technologies, ThermoFisher, Carlsbad, CA, USA). The height and colloid area of the liver epithelium were determined using Image-Pro Plus 6.0. 

### 4.5. Liver Transcriptome Sequencing and Annotation

Liver samples were collected and retained in RNAwait. RNA from the samples of each group was extracted and purified using TRIzol^®^ reagent (Catalogue 12183-555, Invitrogen, ThermoFisher, Carlsbad, CA, USA); RNA quality was determined using an Agilent 2100 bioanalyzer (Agilent, ThermoFisher, Carlsbad, CA, USA), while total RNA was quantified using NanoDrop2000 (Thermo Fisher Scientific, Wilmington, DE, USA). RNA transcriptome libraries were established using the TruSeqTM RNA sample preparation kit (Illumina, ThermoFisher, Carlsbad, CA, USA). LncRNA library construction was prepared for sequencing by mRNA fragment, cDNA synthesis, end repair, adaptor ligation, and polymerase chain reaction (PCR) amplification. The samples were sequenced using an Illumina Novaseq 6000 platform at Shanghai Majorbio Bio-pharm Technology Co., Ltd, (Shanghai, China). Clean sequence data were aligned to the Rattus_norvegicus Rnor_6.0 reference genome (http://www.ensembl.org/Rattus_norvegicus/Info/Index, accessed on 1 November 2022) and assembled using Cufflinks. Expression values were calculated as transcripts per million reads by RNA-seq through expectation maximization. Genes with adjusted *p* ≤ 0.001 and fold change (FC) ≥ 2.0 relative to the CTRL group were identified as differentially expressed genes (DEGs). The coding potential calculator, coding-non-coding index, and coding potential assessment tool were then used to filter transcripts with coding potentials. Reference was also made to the lncRNAs in GREENC (http://greenc.sciencedesigners.com/, accessed on 1 November 2022). The expression level of each lncRNA was calculated according to fragments per kilobase of exon per million mapped reads. Significant DElncRNAs were extracted with |log2FC| > 1 and FDR < 0.05 by EdgeR. Subsequently, Kyoto Encyclopaedia of Genes and Genomes (KEGG) pathway analysis was performed using the online platform I-Sanger (www.i-sanger.com, accessed on 1 November 2022). Canonical pathways, diseases, and regulated molecules were further analyzed through QIAGEN Ingenuity Pathway Analysis. All clean reads were uploaded to the National Center for Biotechnology Information Sequence Read Archive (accession: PRJNA857166).

### 4.6. RNA Isolation and Quantitative Real-Time PCR

TRIzol reagent (Life Technologies, ThermoFisher, Carlsbad, CA, USA) was used to extract total RNA from the samples before being transcribed into cDNA using a first-strand cDNA synthesis kit (TaKaRa, Japan). mRNA levels were then quantified by quantitative PCR (qPCR) on a real-time PCR system using SYBR Premix Ex Taq II (TaKaRa, Japan). The mean of the triplicate cycle thresholds (CTs) of the target gene was normalized to the mean of the triplicate CT of the reference β-actin gene using the formula 2^−ΔΔCt^. Resultantly, relative gene expression levels were attained. Table 1 shows the primer sequences used for real-time RT-PCR analysis.

### 4.7. Statistical Analysis

All experiments were performed in triplicate biological replicates. Statistical analysis was completed using SPSS 22.0; Duncan’s method was used for multiple comparisons; Student’s *t*-test was used to compare the experimental groups and control group. Statistical significance was indicated by a *p* value less than 0.05.

## 5. Conclusions

Our results have revealed the prevention of LPS-induced liver inflammation and oxidative stress following Nar administration. Further, RNA-seq reflects Nar’s ability to regulate differential lncRNAs and related mRNAs, thereby conveying its protective effect against liver injury. Activation of these genes was also shown to have a preventive effect on inflammation in mouse liver AML12 cells and HFD-induced obese mice. Overall, this study highlights a potential new solution for the treatment of liver dysfunction.

## Figures and Tables

**Figure 1 molecules-28-00198-f001:**
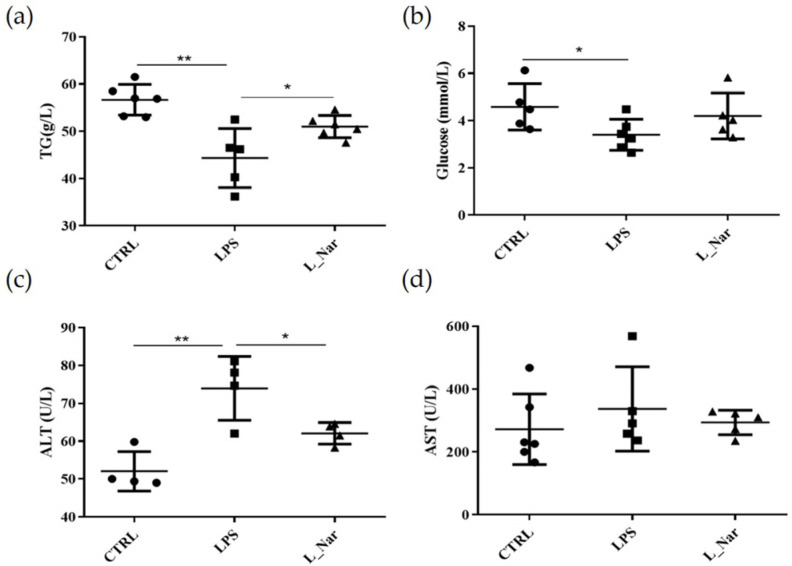
Effects of naringenin on serum parameters of LPS-induced mice: (**a**) serum TG; (**b**) serum glucose; (**c**) ALT; (**d**) AST levels. Data are mean ± SEM. * *p* < 0.05, ** *p* < 0.01 versus CTRL group, LPS group or L_Nar (LPS+Nar) group; TG: triglyceride; ALT: alanine aminotransferase; AST: aspartate aminotransferase.

**Figure 2 molecules-28-00198-f002:**
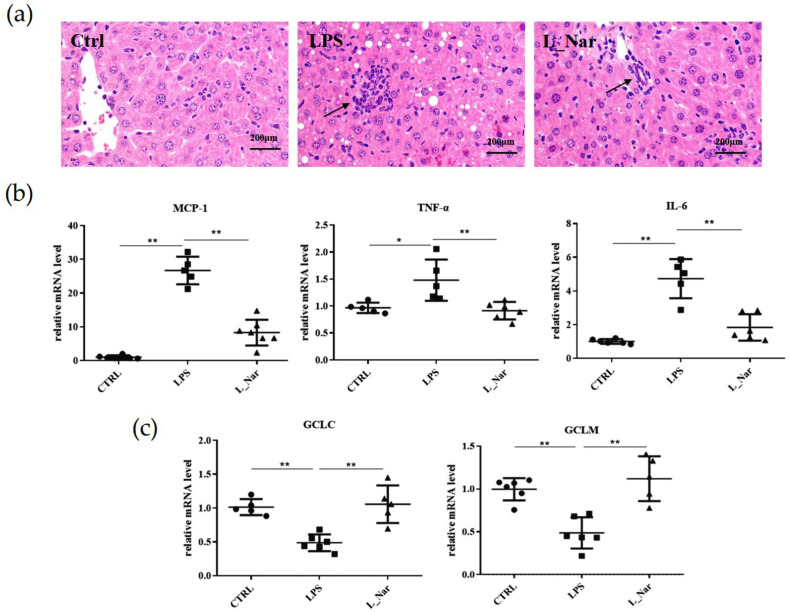
Effect of naringenin on inflammation and oxidative stress in the liver of mice: (**a**) H&E staining of liver sections (magnification, 20x), arrows indicate inflammatory infiltration; (**b**) expression of inflammatory factors MCP1, TNFα, IL-6; (**c**) antioxidant factors GCLC and GCLM mRNA levels. All values are mean ± SE (*n* = 6). * *p* < 0.05, ** *p* < 0.01 versus CTRL group or LPS group; MCP1: mononuclear cell chemotactin-1; TNFα: tumor necrosis factor-α; IL-6: interleukin 6; GCLC: glutamate-cysteine ligase catalytic subunit; GCLM: glutamate-cysteine ligase modifier subunit.

**Figure 3 molecules-28-00198-f003:**
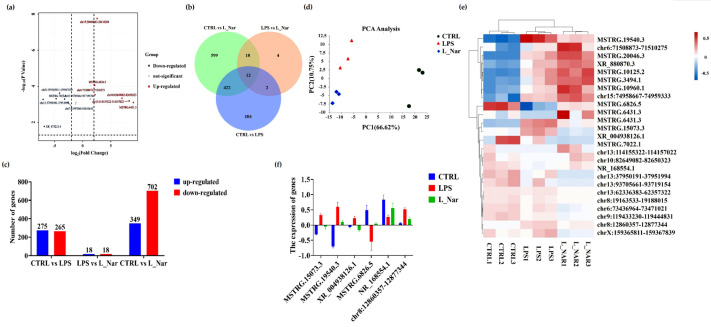
Effects of naringenin on the liver transcriptome of liver tissue of mice: (**a**) volcanic map analysis of DEGs; (**b**) Venn diagrams representing the DEGs specific or common among CTRL, LPS, and L_Nar groups, respectively; (**c**) number of significantly upregulated and downregulated genes in comparisons of the CTRL and LPS groups, LPS and L_Nar groups and CTRL and L_Nar groups; (**d**) principal component analysis; (**e**) hierarchical clustering of the liver samples and the expression pattern of all DEGs based on the transcriptomic profiles; (**f**) expression tendencies of differentially expressed long noncoding RNA between LPS and Nar. Data are expressed as mean ± SEM (*n* = 3). DElncRNA: differential expression lncRNA.

**Figure 4 molecules-28-00198-f004:**
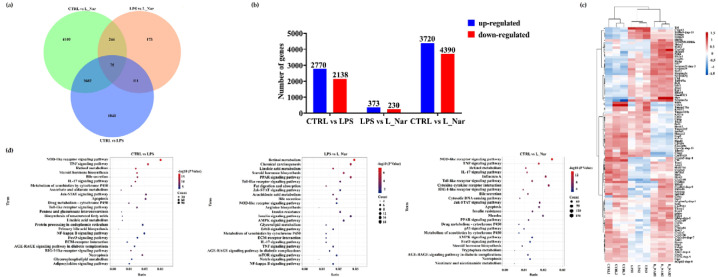
Effects of naringenin on the liver mRNA of mice: (**a**) Venn diagram analysis of co-expressed genes derived from RNA sequencing; (**b**) number of significantly upregulated and downregulated genes in comparisons of the CTRL and LPS groups, LPS and L_Nar groups, and CTRL and L_Nar groups; (**c**) hierarchical clustering of the liver samples and the expression pattern of DEmRNAs based on the transcriptomic profiles; (**d**) comparison of the KEGG pathways enriched for upregulated and downregulated genes. Data are expressed as mean ± SEM (*n* = 3). DEmRNA: differential expression mRNA.

**Figure 5 molecules-28-00198-f005:**
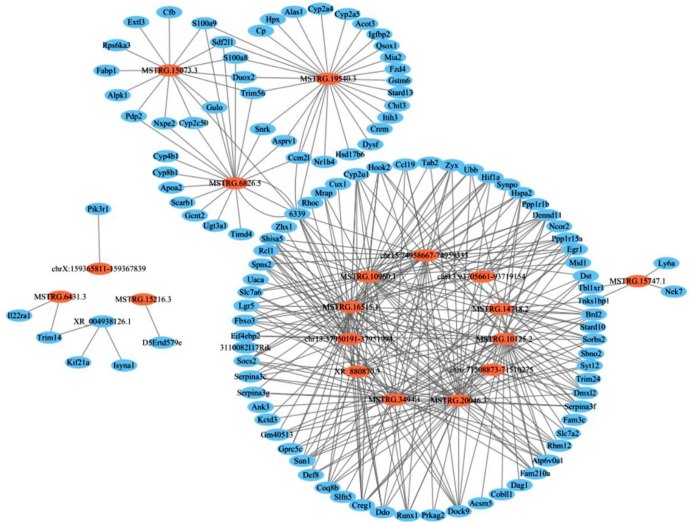
LncRNA (orange) and mRNA (blue) differential gene co-expression networks in the liver.

**Figure 6 molecules-28-00198-f006:**
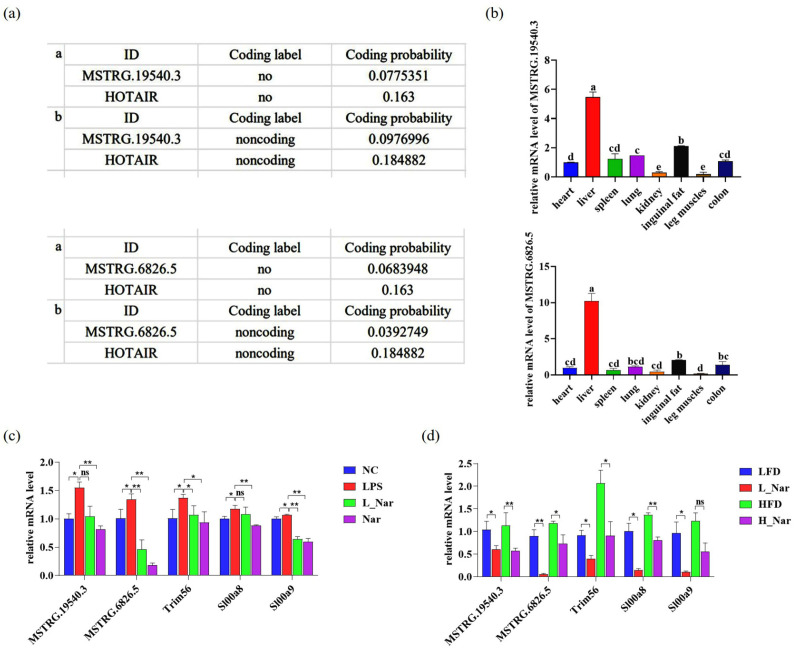
Effect of Nar on lncRNAs and their target genes in AML-12 cells and diet-induced obese mice: (**a**) protein coding ability prediction: a is the CPAT result, b is the CPC result; (**b**) tissue expression profile; (**c**) real-time PCR detection of lncRNAs and related target genes mRNA expression in AML-12 cells and (**d**) diet-induced obese mice. Data are expressed as mean ± SEM (*n* = 3). * *p* < 0.05, ** *p* < 0.01; Bar graphs with the same superscript letters indicate no significant differences (*p* > 0.05), while with different superscript letters indicate significant differences (*p* < 0.05).

**Figure 7 molecules-28-00198-f007:**
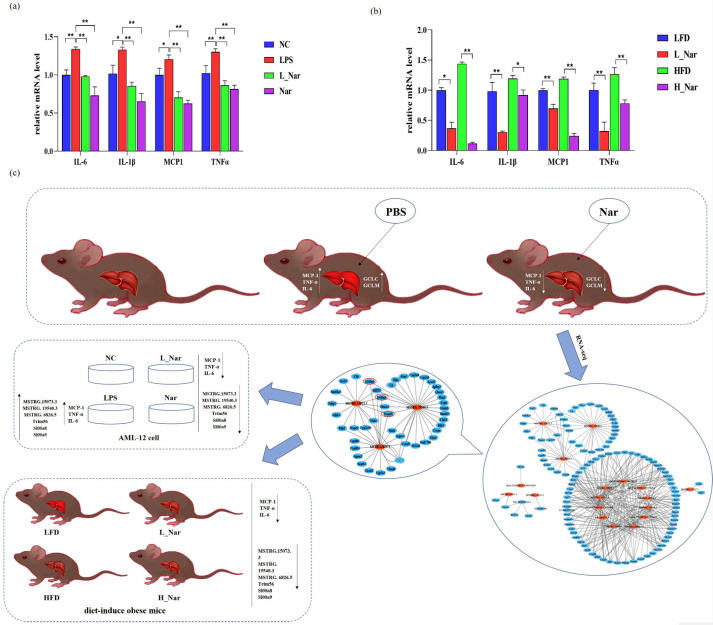
Effects of naringenin on AML-12 and diet-induced inflammation in obese mice: (**a**) mRNA expression of IL-6, IL-1β, MCP1, and TNFα in liver samples of the NC group, LPS group, L_Nar group, and Nar group, respectively; (**b**) real-time PCR detection of inflammatory factors mRNA expression. Data are expressed as mean ± SEM (*n* = 3). * *p* < 0.05, ** *p* < 0.01. (**c**) Schematic diagram of naringenin experiment. IL-1β: interleukin-1β.

**Table 1 molecules-28-00198-t001:** Primers for RT-PCR analysis.

Gene	Forward Primer (5′ → 3′)	Reverse Primer (5′ → 3′)
Glutamate-cysteine ligase catalytic subunit (GCLC)	TCAGCCTCCTCCTCCAAACTCC	TGAGCACACACAAACCAC
Glutamate-cysteine ligase modifier subunit (GCLM)	TCACAATGACCCGAAAGAACTG	ACCCAATCCTGGGCTTCAT
Tumor necrosis factor α (TNFα)	GCACTGAGAGCATGATCCGAGAC	CGACCAGGAGGAAGGAGAAGAGG
Interleukin-6 (IL-6)	ATAAGGGAAATGTCGAGGCTGTGC	GGGTGGTGGCTTTGTCTGGATTC
Monocyte chemoattractant protein-1 (MCP1)	AGCACCAGCCAACTCTCAC	TCTGGACCCATTCCTTCTTG
Interleukin-1β (IL-1β)	CCAATTCAGGGACCCTACCC	GTTTTGGGTGCAGCACTTCAT
MSTRG.19540.3	GGGAAACAGCGCGACCACCC	TGACGCAGGTGTACGCAGATCA
MSTRG.6826.5	ACTAGAGCAAAGAGACAGGACGAAGC	GTGTGTACTGTAGCCACTTAATTCTGT
Calcin-binding protein a9 (S100a9)	ACTGGGCTTACACTGCTCTTACCAA	CCTTTAGACTTGGTTGGGCAGCTGT
Calcin-binding protein a8 (S100a8)	TGTAGACATATCCAGGGACCCAGCC	TCTGGAAGGGAAGAGCGTTGTCTC
Tripartite motif containing 56 (Trim56)	TTGTTTCTCTTGCAGGTAGTCAACT	CAACCCACTCTCTCCTGGTCCCTAA
β-actin	GGCACCACACCTTCTACAATG	GGGGTGTTGAAGGTCTCAAAC

## Data Availability

Not applicable.

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
