# Peer review of "Naringenin Prevents Oxidative Stress and Inflammation in LPS-Induced Liver Injury through the Regulation of LncRNA-mRNA in Male Mice"

_molecules, 2022, doi:10.3390/molecules28010198_

Round 1

Reviewer 1 Report

Your articles provide relevant information about the antioxidant and anti-inflammatory activities of Naringenin. However, the inaccurate choice of words, not clear logic, and the figure fonts are not clear are causes of the decline in the value of the information presented. So, I think this article should be revised by following suggestions.

Specific recommendations

Line 17: The description should be more precise.

Lines 81-82: What does the phrase "opposing effect" mean?

Results

You should precisely use the group name “Nar” or “L_Nar” in the results. For instance, lines 94 and 96 in chapter 2.1, lines 104, 105, and 108 in chapter 2.2, and other chapters in the results section.

In cases when there is a significant change, P-values should be mentioned. Such as lines 89-90 and 93 (2.1), and lines 108 and 110 (2.2).

Lines 109-111: significantly prevented means that better or worse?

Lines 125-128: DEGs changes should be described more precisely.

Line 164: How to screen the 14 differential lncRNA?

Discussion

Lines 207-224: This part appears to be more of an introduction or background section. The first paragraph of the discussion section must describe the answer to your hypothesis.

Line 225: “prevent a decrease” mean that better or worse?

Lines 226-230: The results compared with previously published reports, if not published then search the biochemical principles to justify significant results.

Lines 254-258: The results compared with previously published reports.

Lines 262-271: This part appears to be more of an introduction or background section.

Lines 286-291: The results compared with previously published reports.

Materials and Methods

Line 298: 30mg/kd/d PBS?

Lines 326-327: You should precisely describe the cell density, L_Nar concentration, and process time.

Reviewer 2 Report

The study focuses on the role of naringenin on LPS induced liver injury from a mechanistic perspective. The study is quite fit in the concept of the journal and provides meaningful insight on the molecule of interest; naringenin.

While reading the manuscript, I would like to bring to the attention of the authors some questions and comments and also points in use of English to consider improving the manuscript.

Abstract:

Revise “its role and mechanism of regulating liver dysfunction” to “its mechanism of action in the regulation of liver dysfunction”

Revise “caused significantly increase” to “caused a significant increase”

Introduction:

Revise “necrosis; these responses contribute” to “necrosis, which contribute”

Revise “in treating liver injury” to “in the treatment of liver injury”

Results:

The authors go directly to section 2.1 without any introductory sentence to the results. This should be addressed. An introductory statement would help the reader get into context. For example, initially I could not understand if this was in vivo. In 2.1 there is no mentioning of mice, so this should be cleared.

Revise “Compared with the CTRL group” to “Compared with the control (CTRL) group

Revise lines 89-90: The authors could also state in their text by what %.

Line 91: The authors mention for the first time L_Nar, which I assume is LPS and Nar. This should be explained once here.

The reviewer noticed that the authors designed their study as, control, LPS, and LPS with Nar, however, they did not have control with Nar administration, which most likely would be the same as the control, or differ if Nar has an effect on control too. Although it could be asked to be included the authors may not include it but this has to be justified in their manuscript.

Revise “The results depict a significant” to “The results revealed a significant”

Figure 1: Add here the explanation that L_Nar is LPS+Nar or in the figure instead of L_Nar use LPS+Nar.

Line 121: What is DEGs? Differentially expressed genes? Please clarify.

The authors mention hierarchical clustering. Please specify which distance metric was used and why did you prefer this distance metric or if you checked any other.

Have the authors considered any other multivariate analysis such as grouping with principla component analysis? I think they may get another nice figure of good separation.

Figure 3. Revise the legend “on the liver transcriptome of mice” to “on the transcriptome of liver tissue of mice”.

Section 2.4 Avoid beginning the sentence with “Similarly,” and further down change the word “yielded”. Start with “A comparison of the transcriptomic changes between CTRL and LPS groups revealed …”

Line 166: Revise “shown in Figure 5” to “(Figure 5)”.

Lines 166-167: Why were the two targets of interest? The authors state they were of interest but an explanation would add or how did they come up for them.

Lines 174-175: This was unclear to me. Did the authors do similar analysis in all tissues? Or just focused on the two of interest? If I understood while reading further this was done with qtRT-PCR, however, this can be mentioned here to avoid the ambiguity.

Discussion:

Line 280: Revise “lesser” to “significantly lower”

Sl00a8 is mentioned but without much critical information about what it exactly is and do. Perhaps the authors can elaborate on that.

Materials and Methods:

If I understood the authors used only male rats. Perhaps this should be mentioned specifically in the title maybe too. I have no objection with using one sex as long as this is stated and justified, even if the justification is to keep the study simple and in one sex, which is perfectly fine with me.

Are the authors providing any table with their transcriptomic data? In my opinion, they should share at least a supplementary table of their Excel version of data or meta data used for the clustering.

Overall, I would suggest the authors revise and edit the language in their revised manuscript and address the aforementioned points. The manuscript is sound and has reasoning to support it publication provided that it will be revised. 
